# Histopathological Gastric Cancer Detection on GasHisSDB Dataset Using Deep Ensemble Learning

**DOI:** 10.3390/diagnostics13101793

**Published:** 2023-05-18

**Authors:** Ming Ping Yong, Yan Chai Hum, Khin Wee Lai, Ying Loong Lee, Choon-Hian Goh, Wun-She Yap, Yee Kai Tee

**Affiliations:** 1Lee Kong Chian Faculty of Engineering and Science, Universiti Tunku Abdul Rahman, Kajang 43000, Malaysialeeyingl@utar.edu.my (Y.L.L.);; 2Department of Biomedical Engineering, Faculty of Engineering, Universiti Malaya, Kuala Lumpur 50603, Malaysia

**Keywords:** histopathology, gastric cancer, deep learning, convolutional neural network, transfer learning, ensemble model

## Abstract

Gastric cancer is a leading cause of cancer-related deaths worldwide, underscoring the need for early detection to improve patient survival rates. The current clinical gold standard for detection is histopathological image analysis, but this process is manual, laborious, and time-consuming. As a result, there has been growing interest in developing computer-aided diagnosis to assist pathologists. Deep learning has shown promise in this regard, but each model can only extract a limited number of image features for classification. To overcome this limitation and improve classification performance, this study proposes ensemble models that combine the decisions of several deep learning models. To evaluate the effectiveness of the proposed models, we tested their performance on the publicly available gastric cancer dataset, Gastric Histopathology Sub-size Image Database. Our experimental results showed that the top 5 ensemble model achieved state-of-the-art detection accuracy in all sub-databases, with the highest detection accuracy of 99.20% in the 160 × 160 pixels sub-database. These results demonstrated that ensemble models could extract important features from smaller patch sizes and achieve promising performance. Overall, our proposed work could assist pathologists in detecting gastric cancer through histopathological image analysis and contribute to early gastric cancer detection to improve patient survival rates.

## 1. Introduction

Gastric cancer is one of the most common cancers and leading causes of cancer-related mortality [1]. Gastric cancer is considered a single heterogeneous disease with several histopathologic characteristics [2], where the gastric cancer presents distinct subtype with different histologic appearance, making the detection a non-trivial task. The clinical gold standard of gastric cancer detection is histopathology screening of a biopsy or surgical specimen using a microscope to identify the cancerous features [3]. This is done conventionally by pathologists by manually screening the tissue biopsies, first by using a low magnification factor to search for potential cancerous region(s) with naked eyes. Once a suspicious region is identified, the pathologists will switch to a high magnification factor to analyze the details of the region. During the diagnostic procedure, the pathologists assess the gigapixel-sized whole slide image (WSI) by traversing the WSI to find the small abnormal region of interest (ROI) as described above repeatedly, to make diagnostic decisions.

However, this conventional and manual visual analysis of tissue biopsies by pathologists is extremely laborious, time-consuming, and subjective, where the conclusion drawn by a pathologist can be different from another. The correct analysis of histopathology is highly dependent upon the expertise and experience of the pathologists. This makes the manual histopathological analysis prone to human errors such as misdetection and misdiagnosis, coupled with a shortage of pathologists, leading to long backlogs in the processing of patient cases and consequently increases the likelihood of delayed cancer detection. 

Since most gastric cancers are adenocarcinomas, there are no apparent symptoms in the early stage or may present with non-specific symptoms such as gastric discomfort which are often mistaken as gastric ulcers and gastritis [4]; this causes a delay in the gastric cancer detection. Early detection of gastric cancer is the key factor to reduce mortality [5]. This can be observed in patients with an early gastric cancer diagnosis and detection; they have a survival rate of above 90% [6]. When detected in the late stage, the survival rate reduces substantially to below 30% [7,8].

The limitations of the manual diagnostic workflow lead to the development of computer-aided diagnosis (CAD) to assist pathologists by making the diagnosis more efficient and autonomous. CAD is gaining attention and becoming more accessible nowadays due to the advancement in digital pathology, resulting in slide scanning quality improvement and cost reduction in digital storage [9]. In addition, these systems not only reduce the time and cost of cancer diagnosis but also the inter-pathologist variability in diagnostic decisions [10].

For gastric cancer detection using histopathological images, various CAD techniques have been explored based on classification and segmentation models. Machine learning is the conventional CAD approach used to perform gastric cancer detection. In this approach, the used models extract handcrafted features such as color, texture, and shape features for the detection [11,12,13]. The common machine learning classifiers are support vector machine (SVM), random forest, and Adaboost [14,15,16].

Later, the deep learning approach is introduced to automate feature selection. Many works have reported deep convolutional neural networks (CNN) achieve promising performance in histopathological image classification and segmentation tasks in cancer [17,18,19], metastasis [20,21], and gene mutation [22,23] analysis; some even reported performance comparable to pathologists’ assessment [9,24,25,26,27,28].

However, a good deep learning model requires huge datasets for training to obtain the optimal model parameters. This has particularly affected the use of deep learning in the histopathology image analysis domain due to the exhaustive manual data labeling process of the histopathology images. The data augmentation strategies such as affine transformation, color transformation, and noise addition are generally applied to expand the small histopathology gastric cancer dataset [11,29].

Another commonly used technique to overcome small dataset problems is the transfer learning method. While some researchers proposed their CNNs and trained them from scratch [30,31], others adopted transfer learning to fine-tune pre-trained networks for gastric cancer detection [5,30].

Each pre-trained network has its unique strengths and limitations in extracting features due to variances in their architectures. Although the networks make correct predictions, each pre-trained network may have a different interpretation of an image, and different important features are extracted in the classification process. Utilizing any one of these networks results in the loss of important features that may be extracted by others; this may lead to insufficient important features or information to generate the correct predictions and hence lead to a lower classification performance.

To address the shortcoming in the feature extraction process by the individual pre-trained networks, this work proposed ensemble models for histopathology gastric cancer detection. The ensemble models exploit the strength and overcome the limitation of each pre-trained network by combining decisions of multiple pre-trained networks in the classification process; therefore, they can generate a larger amount of important features or information that are required to make the correct predictions and subsequently improve the classification performance. This is especially useful for the histopathology domains because the WSIs are generally in high resolution; processing them directly would require high computational power; thus, they are commonly cropped into smaller patches with lower resolution for CAD. Although the smaller patches are more computationally friendly, it comes at a cost of lower performance due to less useful features or information in the images. The proposed deep ensemble learning has great potential to overcome this pressing issue in histopathological image analysis.

To verify the effectiveness of our proposed models, their performance was evaluated on the new publicly available gastric cancer dataset Gastric Histopathology Sub-size Image Database (GasHisSDB) [32].

The main contributions of this paper are that (1) deep ensemble learning models that are effective for gastric cancer detection are developed. This is proven by the better performance compared to the state-of-the-art studies on the GasHisSDB dataset; (2) deep ensemble learning can still accurately classify the gastric histopathological images with lower resolution. Therefore, it is feasible to reduce the specifications of the digital scanner, data storage, and high computational server required in the histopathology tasks, potentially translating to higher likelihood of early gastric cancer detection to improve patient survival rate.

## 2. Related Works

The classical machine learning approach based on handcrafted feature extractions was used in automating histopathology tasks initially. Doyle et al. [33] extracted various combinations of handcrafted textural and graph features such as gray level features, Haralick features, Gabor filter features, the Voronoi diagram, Delaunay triangulation, minimum spanning tree, and nuclear features. After that, the authors applied spectral clustering algorithms as dimensionality reduction methods to filter the useful features before passing them to SVM to classify whether the images are normal or breast cancer. The model achieved an accuracy of 95.8% in cancerous image detection and 93.3% in cancer image grading. In the work of Kather et al. [34], six distinct sets of handcrafted texture descriptors including lower-order and higher-order histogram features, local binary patterns, gray-level co-occurrence matrix, Gabor filters, and perception-like features were combined into a feature set; after that, various classifiers including the 1-nearest neighbor, linear SVM, radial-basis function SVM, and decision trees were used for the colorectal image binary and multiclass classification. The proposed work managed to achieve 98.6% accuracy in the binary classification and 87.4% accuracy in the multiclass study. Although the classical machine learning approach can achieve promising performance, it requires in depth expertise in the histopathology domain to design meaningful features, which serve as its shortcoming and barrier to developing an effective machine learning model.

To address this problem, deep learning approach is introduced for histopathology task automation. Unlike machine learning, deep learning models do not require handcrafted features as the input; they can learn the required features automatically. However, a huge dataset is usually needed for the deep learning models to learn the features effectively and then achieve a high performance.

Data augmentation and transfer learning are two common methods used to address the huge dataset requirement in training deep learning models. The previous generates artificial samples to expand the dataset. In the work of Sharma and Mehra [35], the dataset was augmented using flipping, translation, scaling, and rotation technique; Han et al. [36] balanced the dataset using the augmentation methods including intensity change, rotation, and flipping; Joseph et al. [37] applied translation, scaling, flipping, and rotation with constant fill mode to expand the dataset. The model accuracies improved by 2.76–12.28% across various magnifications in [35], 3.4% at the image level and 5.8% at the patient level in [36], and 4.52–8.17% across various magnifications in [37] in the respective tasks after the data augmentation.

The second method to overcome the huge dataset requirement is transfer learning, where a model that has been trained for one task is applied as a starting point of a model to perform a different task. In the work of Al-Haija et al. [38], the pre-trained ResNet50 was fine-tuned for the breast cancer classification task; Mehra [39] compared the transfer learning and training from scratch methods using three models which are VGG-16, VGG-19, and ResNet-50; Celik et al. [40] proposed transfer learning using the pre-trained networks DenseNet-161 and ResNet-50. The pre-trained networks accuracies improved by 5.9–14.76% in [38], 12.67% (between best performing models) in [39], and 1.96–6.73% in [40] in the respective tasks over the custom CNNs or training the models from scratch.

Although the methods above have achieved relatively good performance in the histopathological image analysis, there is another notable method called ensemble learning that can be integrated with these methods to further improve the classification performance. Ensemble learning involves aggregating the output decisions of multiple base models, which would be the pre-trained networks in this case, through relatively simple ensemble strategies to make the final predictions. The intuition behind the ensemble model is that each base model may have its limitation in feature extraction despite its good performance, and these limitations can be overcome through the strength of the other base models. Hence, by combining multiple base models, the ensemble model has a wider coverage of extracted features, resulting in better performance.

For instance, Ghosh et al. [41] proposed an ensemble model concatenating the results of DenseNet-121, InceptionResNetV2, Xception, and custom CNN to classify 112,180 colorectal images, which are resized into 100 × 100 pixels, into multiple classes. Different weights were assigned to the results of each base model depending on their individual performance. The ensemble model ultimately achieved 99.13% balanced accuracy. In the work of Zheng et al. [42], the weighted voting strategy was used as ensemble method to aggregate pre-trained networks including VGG-16, Xception, ResNet-50, and DenseNet-201 in performing breast cancer multiclass classification on 7909 images across four magnifications, achieving accuracy 98.90%. Paladini et al. [43] proposed using the feature concatenation strategy to aggregate the feature outputs of pre-trained networks including ResNet-101, ResNeXt-50, Inception-V3, and DenseNet-161 and consequently processed the aggregated feature vectors through fully connected and classification layers for the colorectal image classification using the dataset consists of 150 × 150 pixels images, achieving an accuracy of 96.16%. The ensemble models accuracies improved by 1.83–2.16% in [41], 0.1–5.25% in [42], and 0.74–2.18% in [43] in the respective tasks over their corresponding base models.

A WSI can be as large as 100,000 × 100,000 pixels; it is costly and time-consuming to annotate the WSI in detail. A common method to process the WSI is to crop it into smaller patches for artificial intelligence training and classification. Downsizing the WSI prior to cropping it into smaller patches is usually conducted for resource constraint centers. This comes at the cost of lower classification performance because the smaller patch size contains less information for classification purposes. Therefore, the selection of patch size demands the consideration of trade-off between computational power and classification performance.

With the promising performance shown by the ensemble models supported by its capability of extracting many important features from multiple base models, the ensemble models have the potential to extract sufficient important features from the smaller patch size yet achieve promising performance. This can have significant impact in making WSI with lower resolution to be more accessible to correct classification by deep learning models, consequently reducing the specification of the digital scanner, data storage, and high computational server required in the histopathology tasks. This would translate to more efficient and autonomous histopathological diagnosis, leading to lower likelihood of delayed cancer detection.

In this study, the GasHisSDB gastric dataset which consists of three sub-databases of patches, being 80 × 80 pixels, 120 × 120 pixels and 160 × 160 pixels, would be used to compare the performance of our proposed ensemble models with the state-of-the-art studies.

## 3. Materials and Methods

### 3.1. Dataset

The GasHisSDB dataset was prepared through a collaboration between Longhua Hospital Shanghai University of Traditional Chinese Medicine, Northeastern University, and Liaoning Cancer Hospital and Institute [32]. The dataset consists of a total of 245,196 patches derived from 600 WSI of 2048 × 2048 pixels. The patches are divided into two classes: normal and abnormal class. The dataset distribution includes 97,076 abnormal image patches and 148,120 normal image patches. The patches were cropped from the WSI at three patch sizes: 80 × 80 pixels, 120 × 120 pixels, and 160 × 160 pixels. These patches of three different sizes were divided into three sub-databases.

The paper applied different algorithms to obtain patches from normal and abnormal tissue regions in WSI, respectively. For the normal tissue region, the patches were directly cropped from the normal pathological section whereas the cancerous tissue region was selected from the abnormal pathological section. Each cropped ROI would have a labeled ground truth. Cancerous image patches with less than 50% cancerous region according to the ground truth map were removed. The WSIs are hematoxylin and eosin (H&E) stained and have a magnification of ×20. The patches were rotated randomly, and dataset orders were scrambled to reduce the correlation between the patches that originated from a similar WSI. The dataset summary and samples are illustrated in Figure 1.

### 3.2. Methodology Overview

This paper proposed CNN architectures based on transfer learning and ensemble models to perform patch binary classification tasks for gastric cancer detection. The workflow is decomposed into 4 stages: (i) preprocessing dataset with empty patch removal and data augmentation, (ii) fine-tuning pre-trained networks or base models, (iii) selecting the best performing base models to combine into ensemble models, and (iv) evaluating and visualizing proposed models using various metrics and class activation map.

### 3.3. Dataset Pre-Processing

Dataset pre-processing can generate a more distributed dataset to assist the subsequent model training for better model performance. The original dataset contains many empty patches that will deteriorate the model’s performance. Therefore, the dataset was first preprocessed by removing empty patches, followed by data augmentation to expand the available data for model training. 

#### 3.3.1. Empty Patch Removal Process

We removed the empty patches from the dataset as these patches are non-informative. The empty patches were defined as patches that contained more than 10% pixels with RGB intensity value more than 230 across all channels. The percentage of removed patches are approximately 4.91% in the 80-pixel sub-database, 4.01% in the 120-pixel sub-database, and 2.86% in the 160-pixel sub-database. The number of samples before and after empty patch removal, respectively, are 146,615 and 139,415 samples in the 80-pixel sub-database, 65,261 and 62,645 samples in the 120-pixel sub-database, and 33,284 and 32,333 samples in the 160-pixel sub-database. After the empty patches were discarded, the rest of the patches were sent for data augmentation.

#### 3.3.2. Data Augmentation

In this process, affine transformation including rotation at an interval of 90°, and horizontal or vertical flipping to the training images was applied to expand the dataset. The generated augmented training set became twice the size of the original training set. The number of samples before and after augmentation, respectively, are 55,766 and 111,532 samples in the 80-pixel sub-database, 25,058 and 50,116 samples in the 120-pixel sub-database, and 12,933 and 25,866 samples in the 160-pixel sub-database.

### 3.4. Pre-Trained Networks as the Base Models

We evaluated several commonly used pre-trained networks, including InceptionV3, Xception, MobileNet, MobileNetV2, DenseNet121, DenseNet169, EfficientNetB0, and EfficientNetB1, to identify and select the best-performing pre-trained networks as base models for the ensemble models in subsequent sections.

#### 3.4.1. InceptionV3

InceptionV3 [44] was a CNN proposed by Szegedy et al. in 2015. This deals with the problem of conventionally increasing model size, leading to too many parameters and computational inefficiency by using factorized convolution. InceptionV3 computes multiple convolutions of different kernel sizes parallelly in a block, contributing to the strength of recognizing features at different scales. InceptionV3 also excels at overcoming vanishing gradient problems using auxiliary classifiers. 

#### 3.4.2. Xception

Xception [45] was a model proposed by Chollet. F. in 2017. This model is inspired by Inception, but it outperformed InceptionV3 on the ImageNet and JFT datasets when using the same number of parameters. Xception adds depthwise separable convolutions and residual connection into the model, with depthwise separable convolutions replacing the Inception modules. The depthwise separable convolutions used in Xception are modified, with pointwise convolution followed by depthwise convolution.

#### 3.4.3. DenseNet Family

DenseNet [46] was a feedforward CNN proposed by Huang et al. in 2017. The model connects feature maps from all preceding layers as the inputs to all subsequent layers in each dense block. The purposes of this architecture are to reduce the vanishing gradient issue, promote feature reuse, and reduce the number of parameters. The feature maps from different layers in a dense block are combined through concatenation, and the downsampling process that cannot be carried out in a dense block for feature concatenation purposes using convolution and pooling operations.

#### 3.4.4. EfficientNet Family

EfficientNet [47] was a model proposed by Tan, M. and Le, Q. in 2019. The main idea is under the resource constraint situation, scaling up models to achieve better accuracy is not feasible; the alternative of balancing the network depth, width, and resolution in the model can be used to improve the performance of the models. The work proposed using the compound scaling method to maintain efficiency in order to achieve better accuracy. This method scales up the baseline model, EfficientNetB0, uniformly across all dimensions to target resource constraints, generating a family of models from EfficientNetB0 to EfficientNetB7.

#### 3.4.5. MobileNet Family

MobileNet [48] was a CNN proposed by Howard et al. in 2017. It is suitable for mobile and embedded vision applications. This model is lightweight and efficient because it has a smaller number of parameters compared to regular CNNs using the depthwise separable convolutions. MobileNetV2 [49] improves upon MobileNet by introducing the inverted residual blocks and bottlenecking features, leading to a significantly lower number of parameters and faster processing time. 

### 3.5. Transfer Learning

Transfer learning using the CNN models mentioned in the previous section were applied. First, their weights were initialized to the weights pre-trained on the ImageNet dataset; then, the models were fine-tuned using the augmented training set.

### 3.6. Ensemble Models Architecture

Three ensemble strategies which are majority voting, unweighted averaging, and weighted averaging were used to build the ensemble models, and their performance was compared. 

For each sub-database, the best-performing base models according to their validation set accuracies were selected to combine into ensemble models. The number of base models was chosen to be either three or five.

The majority voting strategy aggregates the prediction classes from base models through the voting process. The class with the highest votes was selected as the ensemble prediction class. The unweighted averaging strategy aggregates the prediction class probabilities from base models by averaging the base model prediction class probabilities for all classes. The class with the highest ensemble prediction probability was selected as the ensemble prediction class. For the weighted averaging method, unlike the unweighted averaging method, the ensemble prediction probabilities for all classes were averaged using an assigned specific weight for each base model under each class. The weights were optimized using feedforward neural networks in contrast to tuning them manually as hyperparameters.

### 3.7. Experiment Setting

The GasHisSDB dataset was split into train, validation, and test datasets with a train-validation-test ratio of 4:2:4 after the empty patch removal process. The stratified splitting was applied so that the ratio between the two image classes is balanced in all the datasets. The dataset distribution is shown in Table 1.

The pre-trained networks or base models employed were originally trained to classify 1000 classes of images on the ImageNet dataset. To adapt the models for our binary classification task, the original output softmax classification layers which have 1000 nodes were replaced with 2 nodes, corresponding to the binary classes. The remaining parts of the models were not modified.

For training parameters, each network was trained for 30 epochs, and the batch size was set to 20. For the backpropagation setting, the stochastic gradient descent (SGD) optimizer, categorical cross-entropy loss, and default learning rate of 0.01 were used. All layers in the networks including pre-trained layers were unfrozen and set to be trainable. The weight of the models at the epoch with the highest validation accuracy was chosen as the final representation of these models.

To implement the weighted averaging ensemble strategy in the ensemble model section, feedforward neural networks were used. The ensemble probabilities for the binary classes were obtained using two feedforward neural networks. Each network deals with a specific class by processing probabilities of this class from the multiple base models. The network consists of input and output layers only with no hidden layers, and the sigmoid activation function was used. Base model probabilities were fed as the inputs, and ensemble probability was used as the output in each network. The ensemble class probabilities from the two feedforward neural networks were aggregated. The class with the higher ensemble probability was selected as the ensemble prediction class. For model initialization, the model weights were set to zero. The training parameters of these feedforward neural networks were 5 epochs using the Adam optimizer, batch size of 20, and a default learning rate of 0.01. 

### 3.8. Model Evaluation and Visualization

#### 3.8.1. Evaluation Metrics

The metrics used for performance evaluation are accuracy (1), precision (2), recall (3), specificity (4), F1-score (5), area under the curve (AUC), and categorical cross-entropy loss. The abnormal or cancerous patches are labeled as positive samples, and normal or healthy patches are labeled as negative samples. True positive, true negative, false positive, and false negative cases are annotated as *TP*, *TN*, *FP*, and *FN*, respectively.
(1)$$Accuracy=\frac{TP+TN}{TP+TN+FP+FN}$$
(2)$$Precision=\frac{TP}{TP+FP}$$
(3)$$Recall=\frac{TP}{TP+FN}$$
(4)$$Specificity=\frac{TN}{TN+FP}$$
(5)$$F1-score=2\;\times \;\frac{Precision\times Recall}{Precision+Recall}$$

#### 3.8.2. Prediction Visualization

Visualizing model decisions on the images can help us to understand the model behaviors better. In this work, the feature maps of the models were visualized using the Gradient-weighted Class Activation Mapping (Grad-CAM) technique. It is a technique used to produce a coarse localization map using gradients flowing into final convolutional layers in the model to highlight important regions in the images used by each base model to make a classification [50].

The whole processing framework of the proposed ensemble learning models is summarized in Figure 2.

## 4. Results

The accuracy of the different fine-tuned base models on the validation set of the 80-, 120-, and 160-pixel categories is shown in Table 2. EfficientNetB1 had the highest accuracy in the 80-pixel sub-database category, achieving 96.75%; DenseNet169 achieved the highest accuracy (98.21%) in the 120-pixel sub-database, and DenseNet121 was the most accurate network (99.10%) in the 160-pixel sub-database. A more complete performance comparison of fine-tuning base models measured using other evaluation metrics is available in the Appendix A.

The top 3 and 5 fine-tuned base models based on the validation accuracy are listed in Table 3. The top 3 models were the variants of EfficientNet and DenseNet. For the top 5 models, they were variants of EfficientNet, DenseNet, and MobileNet. These were selected as the base models of the deep ensemble learning for the testing set. 

The performance of the different fine-tuned base models and ensemble models on the testing set of the 80-, 120-, and 160-pixel categories is shown in Table 4, Table 5 and Table 6. For the 80-pixel sub-database (Table 4), the best performing base model on the testing set was DenseNet169, achieving an accuracy of 96.67% followed by DenseNet121 and the EfficientNet variants. The ranking was slightly different compared to the performance on the validation set (Table 3), but the performance of the top base models was not too different (less than 1% difference in accuracy). When the top 3 and 5 base models were used for ensemble models according to the selections in Table 3, Ensemble-UA5 had the highest accuracy in the 80-pixel sub-database category, achieving an accuracy of 97.72% and the best AUC, precision, recall, specificity, and F1-score compared to all the tested base and ensemble models.

The performance of the different base models and the ensemble models on the 120-pixel sub-database testing set is presented in Table 5. DenseNet169 again was the best performing base model, achieving an accuracy of 98.17% followed by DenseNet121 and the EfficientNet variants. The ranking of the top 3 and 5 base models was the same in the testing (Table 5) and validation set (Table 3). When the top performing base models were used in the ensemble learning, Ensemble-WA5 achieved the highest accuracy (98.68%) and the best AUC, precision, recall, specificity, and F1-score compared to all the tested base and ensemble models in the 120-pixel sub-database.

In the 160-pixel sub-database testing set (Table 6), DenseNet121 was the best performance base model, obtaining an accuracy of 98.68% followed by the DenseNet169, MobileNetV2, and EfficientNet variants. The ranking of the top 3 and 5 base models was not the same in the testing (Table 6) and validation (Table 3) set, but their performance was close to each other (less than 1% difference in accuracy). When the top 3 and 5 base models were used for ensemble models according to the selections in Table 3, Ensemble-UA5 was the most accurate network, achieving an accuracy of 99.20% and the best AUC, precision, recall, specificity, and F1-score compared to all the tested base and ensemble models in the 160-pixel sub-database.

## 5. Discussion

### 5.1. Performance Analysis of the Base and Ensemble Models

From the performance of the various base and ensemble models in Table 4, Table 5 and Table 6, three interesting trends were observed. Firstly, the ensemble models always outperformed the fine-tuned base models under each sub-database. The lowest base model accuracy (94.56%) was recorded by InceptionV3 in the 80-pixel sub-database whereas the highest accuracy (98.68%) base model was DenseNet121 in the 160-pixel sub-database. Overall, all the base models were able to perform relatively well in gastric cancer detection. 

The Grad-CAM maps highlighting the important regions in the histopathological images used by the top 5 base model to make a classification are plotted in Figure 3. It is interesting to notice that all the base models relied on different regions in the images to classify them. Therefore, when the ensemble models were applied, more features could be analyzed, leading to consistently better results when compared to the individual base model.

The second interesting observation was that the top 5 ensemble model was consistently better than the top 3 ensemble model in each of the sub-databases tested when the same ensemble strategy was used. This was because all the base models were able to perform relatively good (>94.56%) and different base models focused on different important regions in the images for the classification. Thus, when more base models were included in the ensemble learning, higher accuracy was obtained.

The third interesting observation was the best performing model accuracy increased as the patch size increased as shown in Figure 4. The best accuracy of the base model increased from 96.67% in the 80-pixel sub-database to 98.68% in the 160-pixel sub-database. The same was also observed for the ensemble model, increasing from 97.72% to 99.20% when the number of pixels of the sub-databases changed from 80 to 160. This was not surprising given more useful features would be available in the images when the image resolution increased.

### 5.2. Performance Analysis of the Proposed Models and the State-of-the-Art Studies on the GasHisSDB Dataset

Table 7 presents the details of the experiment and the best-performing models from our study and the previous state-of-the-art studies on the GasHisSDB dataset. From the table, all the studies used either 20% or 40% of the data as the testing set. We chose the latter to assess the performance of our proposed models.

All our ensemble models managed to obtain a gastric cancer detection accuracy higher than the reported performances in the literature (Table 4, Table 5 and Table 6), but only our best proposed ensemble model for each of the sub-database are listed in Table 7. The highest performance improvement of our proposed ensemble model when compared to the literature was in the 80-pixel sub-database, where our best obtained detection accuracy was 97.72, approximately 1.44% (97.72–96.28%) higher than the best reported accuracy in the literature; this was followed by 0.74% improvement in the 120-pixel sub-database and 0.37% in the 160-pixel sub-database. As discussed above, as the resolution of the images increased, more useful features would be available in the images; thus, the performance of the best model in the literature would be better, leading to a smaller percentage of improvement when compared to our proposed model.

More comprehensive performance comparison tables, which include performance metrics such as AUC, precision, recall, specificity, and specificity, are available in the Appendix A. These supplementary tables show that our models not only outperformed the previous state-of-the-art studies in terms of the accuracy but also in other metrics.

Our proposed ensemble models managed to obtain the state-of-the-art gastric cancer detection accuracy even with the ratio of training/validation/testing as 40%/20%/40%, meaning we used equal or lesser data for training and validation compared to the reported studies. This performance highlighted the superiority and robustness of our proposed models. The good performance of our work could be attributed to (1) the pre-processing steps taken and (2) the ensemble models used. In our work, the histopathological dataset was first preprocessed by removing empty patches to filter the non-informative images, thus improving the model feature learning. After that, data augmentation was applied to expand the available data for model training. These pre-processing steps were not taken by previous work in the literature. 

As shown by the Grad-CAM maps in Figure 3, each base model focuses on different regions in the image to make a classification. Our proposed ensemble models allow the final decision to rely on wider image features thus resulting in the best gastric cancer detection accuracy on the GasHisSDB dataset. 

### 5.3. Extended Experiments

Our proposed ensemble models were able to achieve the best performance in the binary classification on the GasHisSDB dataset. To prove the effectiveness and robustness of our proposed ensemble models and also to show that the proposed work is not sample/dataset limited, we further experimented these models on a different histopathology dataset named Histology Image Collection Library (HICL) histopathology larynx dataset [53,54]. This is a multi-class dataset which consists of Grade I, II, and III tumors and has a total of 224 images across all three classes. The immunohistochemistry (IHC) stains were applied on the images, different from the GasHisSDB dataset which used H&E stains.

Each image is presented in various resolutions, and we selected the 534 × 400 pixel and 1067 × 800 pixel datasets to perform the extended experiment. Each image was cropped into multiple 200 × 200 pixel patches so that the proposed ensemble models could do patch classification, just like the GasHisSDB dataset. The 534 × 400 pixel images were cropped into 4 patches, and the 1067 × 800 pixel images were cropped into 20 patches. Each patch was directly assigned the class label of its corresponding image. The same preprocessing steps, models, and experiment settings as described in Section 3 were used, except the output softmax layers of the models were set to three nodes instead of two, to cater for the three classes in the extended experiment dataset.

As shown in Table 8, our proposed ensemble models had good generalization ability and achieved the highest overall accuracies of 96.47% using Ensemble-MV5 and 97.99% using Ensemble-WA5 and Ensemble-UA5 for the 534 × 400 pixel and 1067 × 800 pixel datasets, respectively. The results were substantially better than the performance reported by [55]. For the 534 × 400 pixel dataset, the accuracy improvements are 17.3–37.54% and 23.29% for single classes and overall accuracy, respectively. For the 1067 × 800 pixel dataset, the accuracy improvements are 8.54–19.17% and 14.84% for single classes and overall accuracy, respectively.

More complete performance comparison using other evaluation metrics are available in the Appendix A; our proposed ensemble models easily beat the best reported results in the literature. All these demonstrated the ability and generalization of our proposed ensemble models to handle different histopathology datasets of different organ origins, different staining methods, and multi-class classification tasks.

### 5.4. Limitations of Our Proposed Study

Although our proposed ensemble models managed to obtain state-of-the-art results in the main and extended dataset, there are several limitations of the work. Firstly, the computational costs are higher due to multiple base models being required to perform the ensemble learning. This is less of an issue compared to the excellent performance offered by the proposed work because histopathological analysis is still manually done by a pathologist in clinical practice at the moment. In addition, graphical processing units (GPUs) can be used to run the base models in parallel to minimize the processing time needed for ensemble learning.

Secondly, the pre-trained weights are transferred from the ImageNet natural image dataset. These are sub-optimal for the histopathology tasks because the image features of the two datasets are quite different, potentially limiting the base model performance. A possible way to address this issue is to fine-tune the base models on another histopathology dataset prior to the target histopathology dataset. By doing so, the networks would have more optimal initial weights on the target histopathology dataset. This should contribute to an improved histopathological detection. 

## 6. Conclusions

In this paper, deep ensemble learning models based on transfer learning of several pre-trained networks such as MobileNet, DenseNet, EfficientNet, InceptionV3, and Xception were developed for gastric cancer detection. It was found that ensemble learning based on the top 5 base models managed to achieve state-of-the-art detection accuracy; this ranged from 97.72 to 99.20% when the image resolution of the histopathological images changed from 80 × 80 pixels to 160 × 160 pixels. The experimental results demonstrated that ensemble models could extract sufficient important features from a smaller patch size yet achieve promising performance. This could lead to lower specifications of the digital scanner, data storage, and computational server required in the histopathology tasks, leading to faster gastric cancer detection and subsequently higher survival rate. In the future, we plan to use GPUs to speed up the processing and to fine-tune the base models on another histopathology dataset first to obtain more optimal initial weights to further improve the performance of gastric cancer detection.

## Figures and Tables

**Figure 1 diagnostics-13-01793-f001:**
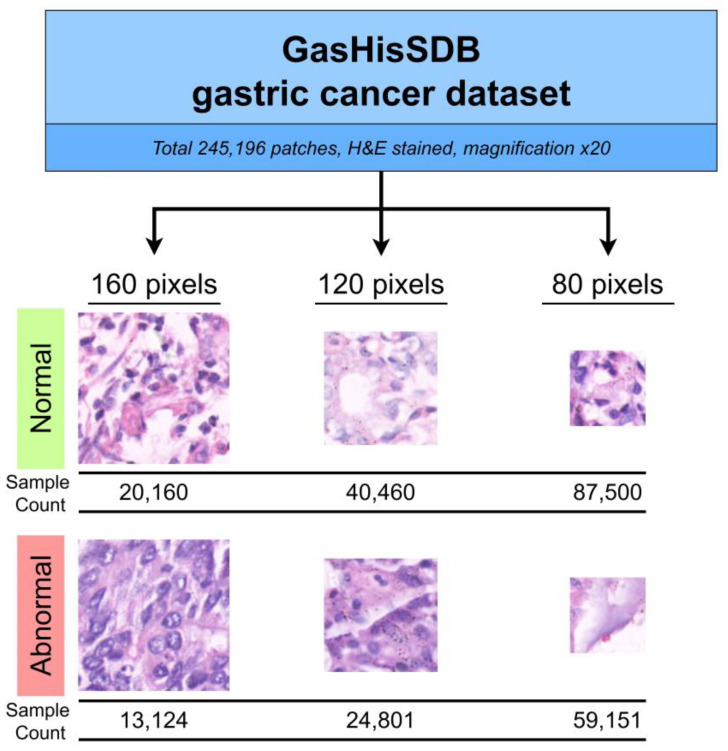
GasHisSDB dataset samples and summary.

**Figure 2 diagnostics-13-01793-f002:**
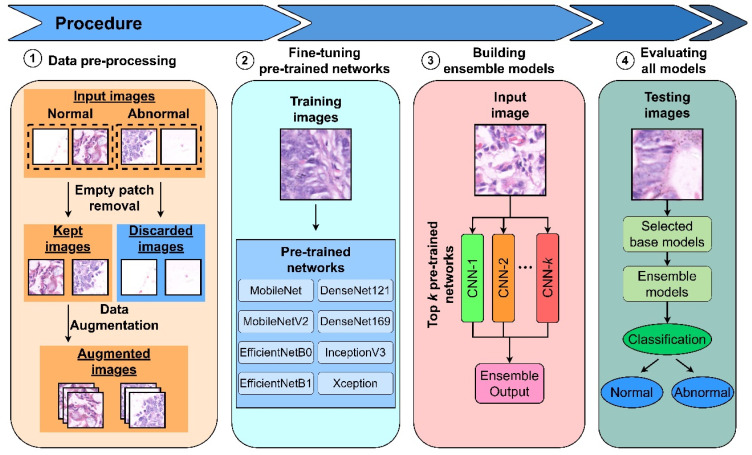
The architecture of the proposed ensemble models.

**Figure 3 diagnostics-13-01793-f003:**
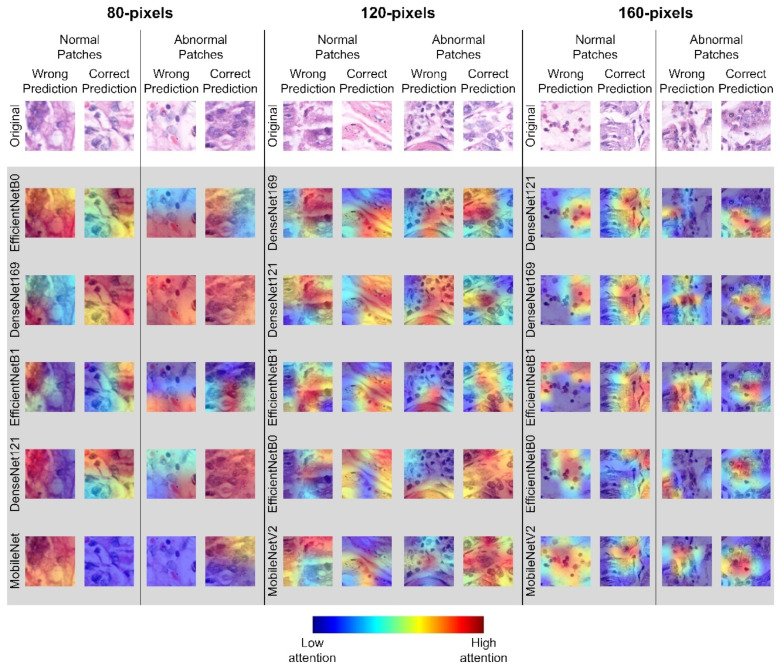
Samples and corresponding Grad-CAM heatmaps generated by the top five base models under all the three sub-databases.

**Figure 4 diagnostics-13-01793-f004:**
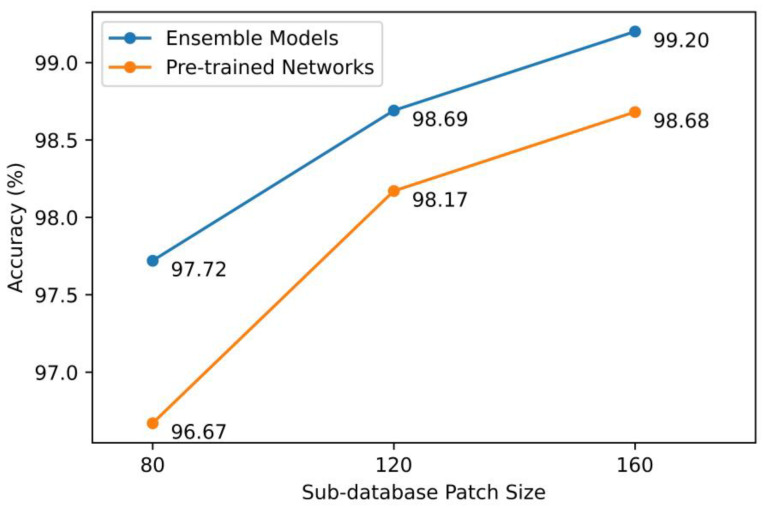
Testing accuracies of best-performing base models and ensemble models across all sub-databases.

**Table 1 diagnostics-13-01793-t001:** GasHisSDB dataset distribution after data pre-processing (empty patch removal and data augmentation).

Database	Number of Samples		
	Augmented Training Set	Validation Set	Testing Set
80-pixels	111,532	27,883	55,766
120-pixels	50,116	12,529	25,058
160-pixels	25,866	6466	12,934

**Table 2 diagnostics-13-01793-t002:** Performance of base models on the 80-, 120-, and 160-pixel sub-database validation sets. The best-achieved results are bold.

Model	Accuracy (%)		
	80-Pixels	120-Pixels	160-Pixels
MobileNet	96.06	97.20	97.99
MobileNetV2	95.49	97.51	98.39
**EfficientNetB0**	**96.75**	97.72	98.48
EfficientNetB1	96.66	97.82	98.50
**DenseNet121**	96.65	98.12	**99.10**
**DenseNet169**	96.73	**98.21**	98.93
InceptionV3	94.75	96.72	98.24
Xception	95.80	97.14	97.79

**Table 3 diagnostics-13-01793-t003:** Ranking of the top 3 and 5 fine-tune base models based on the validation accuracy; they are selected as the base models to be used in the ensemble models.

Ranking	80-Pixels	120-Pixels	160-Pixels
1	EfficientNetB0	DenseNet169	DenseNet121
2	DenseNet169	DenseNet121	DenseNet169
3	EfficientNetB1	EfficientNetB1	EfficientNetB1
4	DenseNet121	EfficientNetB0	EfficientNetB0
5	MobileNet	MobileNetV2	MobileNetV2

**Table 4 diagnostics-13-01793-t004:** Performance of the different deep learning models on the 80-pixel sub-database testing set. The best-achieved results are bold. For the ensemble learning models, WA stands for weighted averaging; UA stands for unweighted averaging, and MV stands for majority voting, and the 3 and 5 at the end of the ensemble models refer to top 3 or 5 base models. All metrics are measured in % unit.

Model	Accuracy	AUC	Precision	Recall	Specificity	F1-Score
MobileNet	95.82	95.73	94.90	95.15	96.30	95.02
MobileNetV2	95.29	94.87	96.36	92.26	97.48	94.27
EfficientNetB0	96.47	96.46	95.26	96.39	96.53	95.82
EfficientNetB1	96.50	96.41	95.83	95.83	96.99	95.83
DenseNet121	96.61	96.30	97.42	94.41	98.20	95.89
**DenseNet169**	**96.67**	**96.70**	**95.26**	**96.88**	**96.52**	**96.07**
InceptionV3	94.56	94.55	92.71	94.47	94.63	93.58
Xception	95.48	95.40	94.34	94.92	95.88	94.63
Ensemble-WA3	97.56	97.51	96.97	97.22	97.80	97.09
Ensemble-WA5	97.69	97.59	97.54	96.95	98.23	97.24
Ensemble-UA3	97.59	97.57	96.80	97.47	97.67	97.13
**Ensemble-UA5**	**97.72**	**97.65**	**97.39**	**97.18**	**98.12**	**97.28**
Ensemble-MV3	97.49	97.47	96.66	97.38	97.57	97.02
Ensemble-MV5	97.66	97.59	97.32	97.10	98.07	97.21

**Table 5 diagnostics-13-01793-t005:** Performance of the different deep learning models on the 120-pixel sub-database testing set. The best-achieved results are bold. For the ensemble learning models, WA stands for weighted averaging; UA stands for unweighted averaging, and MV stands for majority voting, and the 3 and 5 at the end of the ensemble models refer to top 3 or 5 base models. All metrics are measured in % unit.

Model	Accuracy	AUC	Precision	Recall	Specificity	F1-Score
MobileNet	97.12	96.88	96.88	95.76	98.00	96.32
MobileNetV2	97.54	97.59	96.00	97.82	97.35	96.90
EfficientNetB0	97.66	97.66	96.42	97.68	97.64	97.04
EfficientNetB1	97.76	97.67	97.09	97.23	98.10	97.16
DenseNet121	97.87	97.70	97.72	96.86	98.53	97.29
**DenseNet169**	**98.17**	**98.02**	**98.04**	**97.30**	**98.74**	**97.67**
InceptionV3	96.63	96.45	95.80	95.63	97.27	95.71
Xception	97.03	96.86	96.43	96.03	97.69	96.23
Ensemble-WA3	98.52	98.36	98.59	97.63	99.09	98.11
**Ensemble-WA5**	**98.69**	**98.59**	**98.54**	**98.13**	**99.06**	**98.33**
Ensemble-UA3	98.53	98.42	98.36	97.90	98.94	98.13
Ensemble-UA5	98.68	98.61	98.38	98.27	98.95	98.32
Ensemble-MV3	98.47	98.35	98.32	97.78	98.91	98.05
Ensemble-MV5	98.64	98.57	98.32	98.23	98.91	98.27

**Table 6 diagnostics-13-01793-t006:** Performance of the different deep learning models on the 160-pixel sub-database testing set. The best-achieved results are bold. For the ensemble learning models, WA stands for weighted averaging; UA stands for unweighted averaging, and MV stands for majority voting, and the 3 and 5 at the end of the ensemble models refer to top 3 or 5 base models. All metrics are measured in % unit.

Model	Accuracy	AUC	Precision	Recall	Specificity	F1-Score
MobileNet	98.00	97.72	98.75	96.28	99.17	97.50
MobileNetV2	98.42	98.36	98.03	98.05	98.66	98.04
EfficientNetB0	98.33	98.29	97.83	98.05	98.52	97.94
EfficientNetB1	98.36	98.32	97.85	98.11	98.53	97.98
**DenseNet121**	**98.68**	**98.58**	**98.66**	**98.07**	**99.09**	**98.36**
DenseNet169	98.57	98.36	99.20	97.25	99.47	98.22
InceptionV3	97.85	97.83	97.02	97.69	97.96	97.36
Xception	97.43	97.34	96.74	96.91	97.78	96.82
Ensemble-WA3	98.94	98.78	99.44	97.94	99.62	98.68
Ensemble-WA5	99.16	99.09	99.19	98.72	99.45	98.96
Ensemble-UA3	99.03	98.93	99.13	98.45	99.42	98.79
**Ensemble-UA5**	**99.20**	**99.14**	**99.23**	**98.80**	**99.48**	**99.01**
Ensemble-MV3	98.97	98.88	99.06	98.40	99.36	98.73
Ensemble-MV5	99.13	99.07	99.10	98.76	99.39	98.93

**Table 7 diagnostics-13-01793-t007:** Performance comparison of our work and the previous state-of-the-art studies on the GasHisSDB dataset. The best-achieved results are bold. Only our best proposed ensemble models are included in this table, but all the proposed ensemble models achieve detection accuracy higher than the performance reported in the literature.

Paper	Training /Validation/ Testing	Dataset Pre-Processing	Model Details	Accuracy (%)		
				80-Pixels	120-Pixels	160-Pixels
[32]	40%/40%/20%	-	VGG16	96.12	96.47	95.90
			ResNet50	96.09	95.94	96.09
[51]	40%/20%/40%	-	InceptionV3 trained from scratch	-	-	98.83 ± 0.05
			InceptionV3 + ResNet50 (feature concatenation)	-	-	98.80 ± 0.12
[52]	60%/20%/20%	-	Local-global feature fuse network	-	-	96.81
[4]	80%/-/20%	-	MCLNet based on ShuffleNetV2	96.28	97.95	97.85
**Our study** **(only the best model is listed)**	40%/20%/40%	Data augmentation, empty patch removal	EfficientNetB0 + EfficientNetB1+ DenseNet121 + DenseNet169 + MobileNet (unweighted averaging)	**97.72**	98.68	**99.20**
			EfficientNetB0 + EfficientNetB1+ DenseNet121 + DenseNet169 + MobileNetV2 (weighted averaging)	97.69	**98.69**	99.16

**Table 8 diagnostics-13-01793-t008:** Performance of the different deep learning models on the 534 × 400 and 1067 × 800 pixel HICL Larynx datasets. The best-achieved results are bold. For the ensemble learning models, WA stands for weighted averaging; UA stands for unweighted averaging, and MV stands for majority voting, and the 5 at the end of the ensemble models refer to the top 5 base models.

		534 × 400 Pixels Dataset				1067 × 800 Pixels Dataset			
Paper	Model	Single Class Accuracy (%)			Accuracy (%)	Single Class Accuracy (%)			Accuracy (%)
		G1	G2	G3		G1	G2	G3	
[55]	LPCANet	81.18	74.46	60.42	73.18	81.30	89.40	78.50	83.15
**Our models**	**Ensemble-WA5**	98.48	89.09	95.92	94.71	**98.28**	**97.94**	**97.67**	**97.99**
	**Ensemble-UA5**	98.48	89.09	95.92	94.71	**98.28**	**97.94**	**97.67**	**97.99**
	**Ensemble-MV5**	**98.48**	**92.73**	**97.96**	**96.47**	97.99	97.94	97.67	97.88

## Data Availability

The GasHisSDB dataset is openly available at https://gitee.com/neuhwm/GasHisSDB (accessed on 13 May 2023). The HICL dataset is available upon request at http://medisp.bme.teiath.gr/hicl/ (accessed on 13 May 2023).

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
