# Peer review of "Histopathological Gastric Cancer Detection on GasHisSDB Dataset Using Deep Ensemble Learning"

_diagnostics, 2023, doi:10.3390/diagnostics13101793_

Round 1
Reviewer 1 Report
In this manuscript, the authors proposed ensemble models that combine the decisions of several deep-learning models. To evaluate the effectiveness of the proposed models, the authors tested their performance on the publicly available gastric cancer dataset, Gastric Histopathology Sub-size Image Database. Their experimental findings showed that the top-5 ensemble model achieved state-of-the-art detection accuracy in all sub-databases, with the highest detection accuracy of 99.20% in the 160 x160 pixels sub-database.
I have minor comments before it gets considered for publication.
- The authors should justify the motivation of this study.
- The proposed methods have little novelty. The authors are encouraged to state their contribution clearly and how it differs from the state-of-the-art methods.
- Why have they selected these models? Did the authors try to use the vision transformers?
- Require broad ablation study.
- The English language should be improved.
Moderate English correction is required.
Author Response
We thank the reviewers for their considered reading of the manuscript and the insightful comments and corrections offered. We have endeavoured to incorporate all these into this revised submission and hoped that the manuscript is now considered suitable for publication in the Diagnostics. Please find our detailed response to each of the reviewers’ comments in the attachment.
Yours,
YKT
On behalf of all authors

Reviewer 2 Report
The manuscript describes an ensemble classification method applied on the GasHisSDB dataset.
The method of defining ensemble classifiers is well-known in the literature, also, the fact that these models outperform specific state-of-the-art classifiers is also well documented. Therefore, while it is possible that this method was never applied to the given dataset, the method is well-known, not novel at all.
On the other hand, the results are presented nicely, the tables are very well edited, results are evaluated thoroughly.
Author Response

(The authors gave the same response as above.)

Reviewer 3 Report
The current paper presents a methodology for gastric cancer detection based on Histopathology images. This methodology is based on ensemble learning, assuming to combine CNN techniques at decision level, by employing majority voting, average voting and weighted average voting. The experiments are performed on a specific dataset, taking into account different resolutions of the input patches. Comparisons with the already existing state of the art results are also performed and the limitations of the current work are discussed as well. The article is generally well written, well organized, demonstrating an increased technical and scientific level, however, the following observations should be considered:
(1.) The state of the art should be presented in a more extensive manner. The contributions of the current paper should be more clearly emphasized and the originality of the current work with respect to the state of the art should be more clearly highlighted.
(3.) Concerning the experimental dataset, the number of patients/class should be specified.
(2.) Regarding the comparisons with other state of the art approaches, more details should be provided and the authors should clarify as well weather the comparisons are performed on the same dataset or not. All the performance metrics, not only the accuracy, should be provided for this comparison.
The quality of the English language is generally good, however, there are some grammar errors within the text and some phrases should be rephrased, especially at the beginning of the "Related Works" section.
Regarding the grammar errors: for example, on page 3, the phrase "deep learning models does not require handcrafted features as the input" should be replaced by "deep learning models do not require handcrafted features as the input".
Author Response
We thank the reviewers for their considered reading of the manuscript and the insightful comments and corrections offered. We have endeavoured to incorporate all these into this revised submission and hoped that the manuscript is now considered suitable for publication in the Diagnostics. Please find our detailed response to each of the reviewers’ comments, annotated changes in manuscript and supplementary materials in the attachments.
Yours,
YKT
On behalf of all authors

Round 2
Reviewer 2 Report
Manuscript have not improved significantly since last submission.
Author Response

(The authors gave the same response as above.)
